# A Multi-Task Benchmark for Korean Legal Language Understanding and Judgement Prediction

**Wonseok Hwang**[a]    **Dongjun Lee**[a]    **Kyoungyeon Cho**[a]    **Hanuhl Lee**[a]    **Minjoon Seo**[a,b]

[a]LBox
[b]KAIST

{wonseok.hwang, dongjun.lee, kycho, leehanuhl}@lbox.kr
minjoon@kaist.ac.kr

## Abstract

The recent advances of deep learning have dramatically changed how machine learning, especially in the domain of natural language processing, can be applied to legal domain. However, this shift to the data-driven approaches calls for larger and more diverse datasets, which are nevertheless still small in number, especially in non-English languages. Here we present the first large-scale benchmark of Korean legal AI datasets, LBOX OPEN, that consists of one legal corpus, two classification tasks, two legal judgement prediction (LJP) tasks, and one summarization task. The legal corpus consists of 147k Korean precedents (259M tokens), of which 63k are sentenced in last 4 years and 96k are from the first and the second level courts in which factual issues are reviewed. The two classification tasks are case names (11.3k) and statutes (2.8k) prediction from the factual description of individual cases. The LJP tasks consist of (1) 10.5k criminal examples where the model is asked to predict fine amount, imprisonment with labor, and imprisonment without labor ranges for the given facts, and (2) 4.7k civil examples where the inputs are facts and claim for relief and outputs are the degrees of claim acceptance. The summarization task consists of the Supreme Court precedents and the corresponding summaries (20k). We also release realistic variants of the datasets by extending the domain (1) to infrequent case categories in case name (31k examples) and statute (17.7k) classification tasks, and (2) to long input sequences in the summarization task (51k). Finally, we release LCUBE, the first Korean legal language model trained on the legal corpus from this study. Given the uniqueness of the Law of South Korea and the diversity of the legal tasks covered in this work, we believe that LBOX OPEN contributes to the multilinguality of global legal research. LBOX OPEN and LCUBE will be publicly available[1].

## 1 Introduction

The application of artificial intelligence in a legal domain has a long history, dating back to 1980s (Ashley, 2017). Although previous legal expert systems based on rules and domain knowledges showed some useful results on certain legal areas such as settlement decisions of product liability disputes (Waterman and Peterson, 1981), such system could not be easily extended beyond narrow domains. More recently, the advances in deep learning are fundamentally changing how and where machine learning can be applied in legal domain, such as legal judgement prediction (Chalkidis et al., 2019; Ma et al., 2021; Zhong et al., 2020a; Xiao et al., 2018), legal content generation (Wu et al., 2020; Tonguz et al., 2021), legal text classification (Chalkidis et al., 2021; Chalkidis and Søgaard, 2022; Hendrycks et al., 2021), legal event detection (Yao et al., 2022), legal information extraction

---

[1]https://github.com/lbox-kr/lbox-open

(Hong et al., 2021), legal contract review (Hendrycks et al., 2021; Koreeda and Manning, 2021), and question answering (Khazaeli et al., 2021; Zhong et al., 2020b; Duan et al., 2019).

This shift of the paradigm to data-driven approaches necessitates the use of large-scale datasets for training machine learning systems. In response to this, legal research groups worldwide have worked on various legal AI datasets across multiple languages recently (Paul et al., 2022; Kapoor et al., 2022; Yao et al., 2022; Glaser et al., 2021; Niklaus et al., 2021; Chalkidis et al., 2022a; Rossi et al., 2021; Chalkidis et al., 2022b; Louis and Spanakis, 2022; Rabelo et al., 2020).

However, despite the fact that Korea has a legal industry market size of ∼4.8B USD[2] it still lacks large-scale corpora, benchmark datasets, and language models in the domain of Korean Law. AI-hub, an organization run by Korean government, released some Korean legal AI datasets in 2020[3] but they consist of a relatively small corpus (obtained from 6k precedents) and a single named entity recognition task, which lack both the scale and the diversity of the tasks.

In this work, we release LBOX OPEN, the first large-scale Korean legal AI benchmark that consists of six datasets: (1) a large-scale legal precedent corpus (PRECEDENT CORPUS), (2) two classification tasks (CASE NAME, STATUTE), (3) two legal judgement prediction tasks (LJP-CRIMINAL, LJP-CIVIL), and (4) one summarization task (SUMMARIZATION). PRECEDENT CORPUS consists of 147k precedents (259M tokens) of which 63k are sentenced in last 4 years and 96k are from the first and the second level courts. CASE NAME consists of 11.3k facts and case name pairs and STATUTE includes 2.8k facts and the corresponding statutes pairs. In both tasks, a model needs to use the facts to predict the corresponding case names or statutes. We also release the extended datasets CASE NAME+ (31k), and STATUTE+ (17.7k) by including infrequent case categories. LJP-CRIMINAL consists of three subtasks: fine amount, imprisonment with labor or imprisonment without labor range predictions. In each task, a model is expected to use the facts to predict the judgement results. The dataset consists of 11k examples from 7 case categories. In LJP-CIVIL, a model is asked to predict the claim acceptance degree for the given facts and the gist of the claim. The range is determined based on how much claimed money from plaintiffs is approved by the judge. SUMMARIZATION consists of 20k precedents from the Korean Supreme Court and the corresponding summaries. In addition, we also release SUMMARIZATION+ (51k) by including examples with long sequences. The comparison of LBOX OPEN to previous studies are presented in Appendix A.1.

We also release the first large-scale Korean legal language model LCUBE (124M, 345M) pre-trained using LBOX OPEN. Comparisons with other language models on our benchmark affirm that a domain-specific corpus is necessary to achieve competent performance on the tasks. For the text classification tasks, LCUBE, which is a decoder-only model based on GPT-2, shows comparable performance with mt5 (Raffel et al., 2020), a competitive encoder-decoder language model with larger size. On the other hand, on the summarization task, LCUBE does not seem to have advantage over other models.

Given the uniqueness of the Law of South Korea and the diversity of the legal tasks covered in this work, we believe that LBOX OPEN not only contributes to the multilinguality of global legal research but also serves as a good example for creating a benchmark for legal language understanding and judgement prediction in other languages. Both LBOX OPEN and LCUBE will be publicly available.

## 2 Background

### 2.1 Korean legal system

Korean court system is based on the three-tiered justice system, which is composed of district courts, the high courts, and the Supreme Court. The first two trials debate both the factual and legal issues of cases whereas the final trial only discusses the legal issues.

Korean legal system is rooted in civil law system. Unlike the common law system, which is widely accepted in the U.S. and the U.K., codified statute are of primary importance in civil law system and precedents only play supplementary roles. Therefore the doctrine of stare decisis, meaning that courts should adhere to precedent in making their decisions, is not institutionally adopted in Korea. Court Organization Act states that a higher court's decision on a matter only binds the lower court on the

---

[2] the size is estimated based on the value-added tax base declaration amount `https://m.lawtimes.co.kr/Content/Article?serial=163743)`

[3] `https://aihub.or.kr/aidata/7974`

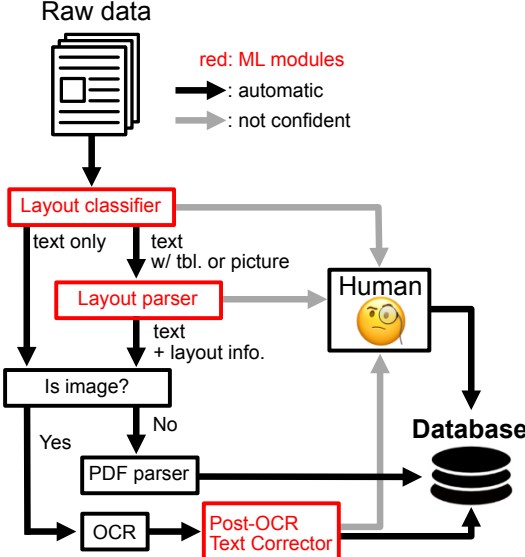

Figure 1: Precedents structuring pipeline. The raw data is formatted as images or PDF.

particular matter[4]. In practice, however, higher courts' decisions, especially those of the Supreme Court, are typically followed by lower courts as statutes alone cannot cover whole complex social phenomena.

## 2.2 Korean precedent

**Structure of Korean precedent**    Korean precedents consist of five main parts, (1) meta information (such as case number, plaintiffs, defendants, and sentencing date) (2) gist of claim from plaintiffs in a civil case, (3) ruling, and (4) reasoning section where (4-1) facts, (4-2) claims, (4-3) reasoning, and (4-4) decisions are loosely separated. The example of precedent is shown in Appendix A.4.

**The redaction process**    The Korean government follows the official redaction process to avoid a possible risk of invading privacy. By the redaction process, any data that may reveal personal information are anonymized before being open to public. Data subjected to anonymization are described in Appendix A.2.

**Precedent disclosure status**    According to Civil and Criminal Procedure Act, the courts are making their judgement decisions public via online service[5] provided by the Supreme Court of Korea. Also the Korean precedents are officially not protected by the copyright law[6]. Despite this, the accessibility of the Korean precedents to the general public is still contentious in that the courts only provide final decisions that are irrevocable while charging a 1,000 won fee per a document. The more discussions on this topic is presented in Appendix A.3.

## 3   LBox Open datasets

### 3.1   Structuring raw data

Korean precedents have been released by the Korean government in two formats: (1) document images, or (2) PDF. In each page, figures and tables are mixed with plain text, which makes parsing non-trivial. To automatically process them, we built our custom data engineering pipeline (Fig. 1). To process raw data, we first use `Layout classifier` based on ResNet (He et al., 2016) that

---

[4]Article 8, Korean Court Organization Act, `https://elaw.klri.re.kr/kor_service/lawView.do?hseq=55374&lang=ENG`

[5]https://www.scourt.go.kr/portal/information/finalruling/guide/index.html

[6]COPYRIGHT ACT, Article 7, `https://law.go.kr/LSW/lsInfoP.do?lsiSeq=192474&viewCls=engLsInfoR&urlMode=engLsInfoR#0000`, `https://law.go.kr/법령/저작권법/제7조`

Table 1: Comparison of precedent corpus. `glaw` indicates the statistics of the set of precedents we retrieved from LAW OPEN DATA.

| Name | LBOX OPEN | glaw | ai-hub |
|---|---|---|---|
| # of precedents | 146,791 | 82,258 | 5,958 |
| # of tokens[†] | 259M | 157M | 24M |
| # of trials from first and second level courts | 56,557 (1st), 39,570 (2nd) | 11,278 (1st), 24,085 (2nd) | 2,218 (1st, 2nd) |
| # of precedents sentenced in last 4 years (2018–2021) | 63,439 | 4,416 | 692 |
| # of precedents of each case type | 58,932 (civil), 54,624 (criminal), 25,007 (administration), 3,266 (patent), 1,237 (family suit), 3,725 (etc) | 36,546 (civil), 18,912 (criminal), 20,824 (administration), 3,219 (patent), 1,210 (family suit), 1,547 (etc) | - |
| # of major case categories[*] | 1,161 | 766 | 59 |

† Based on KLUE (Park et al., 2021) tokenizer.
∗ The categories with at least 10 precedents in the corpus.

classifies each page into `text only` or `text w/ tbl or pictures`. `Layout classifier` was trained using 300k training examples consisting of document image and corresponding label pairs. Next, we use `Layout parser` that is based on Mask-R-CNN (He et al., 2017; Wu et al., 2019; Zhong et al., 2019) to segment any non-textual component from pages. Next, if the page is in PDF format, we extract text and use a custom rule-based parser before sending them to the database. When the page is in image format, we use a proprietary OCR engine[7] to extract text segments and their coordinates. Then we use our custom language model to correct OCR errors before sending it to the database. The model uses the Korean character level transformer as backbone [8]. When the confidence scores from the ML modules are smaller than certain thresholds, corresponding pages are processed manually. Final computer readable documents are saved in JSON format that includes (1) meta information such as case name, sentencing dates, and names of attendees, (2) ruling, (3) gist of claim, (4) appeal, (5) reasoning body that consists of facts, claims, reasoning, and decision of judges. The further details about the data engineering pipeline are explained in Appendix A.6.

## 3.2 Datasets

**PRECEDENT CORPUS** The AI hub, an organization run by the Korean government, released a legal corpus in 2020. Although this has been a good stepping stone for legal AI research, it consists of only 6k precedents. As approximately 1.5 million Korean precedents are created every year[9], it is difficult to expect the released precedents can cover various legal activities. To subside this limitation of the previous work, we release PRECEDENT CORPUS consisting of 147k precedents (259M tokens) that includes 96k precedents from the first and second level trials and 63k recent precedents. We retrieve 82k Korean precedents raw data from LAW OPEN DATA[10] and additional 65k precedents from our internal database. The 57% of precedents retrieved from LAW OPEN DATA consist of the trials of the Supreme Court where the factual issues of cases are not discussed. As statutes cannot comprehensively cover diverse aspects of complex social phenomena, precedents are used to cover the empty spaces of the statutes and thus the precedents from the first and the second trials discussing factual issues are most important to legal practitioners. In this regard we include additional 65k precedents from our internal database where 94% of them are from first and second trials from recent 4 years (2018-2021). The difference between PRECEDENT CORPUS and other corpus is summarized in Table 1.

**CASE NAME** Automatic categorization of legal documents or legal questions is often necessary for various downstream tasks. For instance, for the given factual description of cases from individuals customers one can consider building a lawyer recommendation system based on the predicted case categories. In this regard, we make CASE NAME dataset that consists of 10k facts and case name

---

[7]We use a combination of Google's (`https://cloud.google.com/vision`) and NAVER Clova's (`https://clova.ai/ocr`) OCR APIs.

[8]`https://github.com/monologg/KoCharELECTRA`

[9]`https://www.scourt.go.kr/portal/justicesta/JusticestaListAction.work?gubun=10`

[10]`https://www.law.go.kr/LSO/main.do`

Table 2: Task overview.

| Name | Task Type | Input | Output | # of case categories | # of classes | Size train:valid:test:test2* |
|---|---|---|---|---|---|---|
| CASE NAME | Classification | facts | case name | 100 | 100 | 8,000:1,000:1,000:1,294 |
| CASE NAME+ | " | " | " | 603 | 603 | 22,494:3,999:-:4,790 |
| STATUTE | Classification | facts | statutes | 46 | 188 | 2,208:276:276:538 |
| STATUTE+ | " | " | " | 434 | 1,015 | 13,317:2,276:-:2,137 |
| LJP-CRIMINAL | Legal judgement prediction | facts | fine, imprisonment w/ labor, imprisonment w/o labor | 7 | $(5, 6, 6)^a$ or $(3, 3, 3)^b$ | 8,400:1,050:1,050:928 |
| LJP-CIVIL | Legal judgement prediction | facts, gist of claim | the claim acceptance degree | 4 | 3 or $13^b$ | 3,742:467:467:403 |
| SUMMARIZATION | Summarization | ruling reasoning section | summary | | - | 16,000:2,000:2,000:- |
| SUMMARIZATION+ | " | " | " | | - | 40,892:5,111:5,111:- |

∗ An additional test set made from precedents not included in PRECEDENT CORPUS. Built for the fair comparison.
a LJP-CRIMINAL consists of three subtasks fine, imprisonment with labor, and imprisonment without labor.
b The number of classes under additional quantization scheme. See Table 7 in Appendix for details.

pairs extracted from first level trials in PRECEDENT CORPUS. In the task, a model is asked to predict case name (case category) for the given facts. We collect 11,294 examples from 100 most frequent case categories such as "fraud" (사기), "drunk driving" (도로교통법위반(음주운전)), and "Violation of the Labor Standards Act" (근로기준법위반) where each category includes 100 examples (Table 2). An example is shown in Appendix A.5.2. We also provide CASE NAME+, a dataset that extends CASE NAME by including infrequent case categories. CASE NAME+ consists of 31,283 examples with total 603 case categories. The data distribution is shown Fig. 2a in Appendix.

**STATUTE** In criminal cases, the ranges of the punishment are determined by corresponding statutes following "no penalty without a law" principle. Thus predicting the statutes for the given facts is the first step to determine legal judgements. For this task, we make STATUTE that consists of 3,298 facts and corresponding statutes pairs. The dataset consists of 46 frequent case categories where each class includes 60 examples (Table 2, Appendix A.5.3). The dataset is extracted from the same precedents used in CASE NAME. We also release STATUTE+, a dataset that extends STATUTE by including less frequent case categories. STATUTE+ includes 17,730 examples with total 434 case categories and 1,015 statutes. The data distribution is shown Fig. 2b in Appendix.

**LJP-CRIMINAL** In criminal cases, judges decide the ranges of punishment based on the facts. In this regard, we prepare LJP-CRIMINAL dataset that consists of 10,500 facts and the corresponding three kinds punishment: (1) fine, (2) imprisonment with labor, and (3) imprisonment without labor (Table 2). In this task, a model needs to predict the punishment from the given facts. We design the task with varying degree of difficulties. At the level 0, a model predicts only the type of punishment if any. At level 1, a model is asked to classify the degree of punishment into three levels null, low, and high. The boundary of low and high is determined in a way that balances the data distribution between two classes (Table. 7 in Appendix, 3rd and 4th rows). At level 2, we further split the range of punishment into five (fine) and six (imprisonment) empirically based on the data distribution and the certain characteristics of Korean legal system. For instance in the fine prediction subtask, KRW 1,000,000/3,000,000 is selected as a boundary as this is the lower bound where public officials can lose their position if found guilty related to sexual/general crime[11]. At level 3, a model predicts the exact numbers and the task becomes regression. The dataset is extracted from the first level criminal trials from PRECEDENT CORPUS with the following case categories: "indecent act by compulsion" (강제추행), "obstruction of performance of official duties" (공무집행방해), "bodily injuries from traffic accident" (교통사고처리특례법위반(치상)), "drunk driving" (도로교통법위반(음주운전)), "fraud" (사기), "inflicting bodily injuries" (상해), and "violence" (폭행). Each category includes 1,500 examples. Only trials with a single defendant and a single charge are selected.

---

[11]Although there is an official sentencing guideline from the sentencing commission of the Korean government (https://sc.scourt.go.kr/sc/engsc/pdf/SentencingGuidelines2021.pdf), the guide provides only the ranges of the possible punishments divided by three high levels–reduced, base, weighted–for individual case categories. Finding the quantization boundaries that are more theoretically grounded is currently under study.

**LJP-CIVIL**    In civil trials, plaintiffs often claim money as a part of compensation and the judges approve a certain portion of the claimed money based on the facts. For this legal judgement procedure, we built LJP-CIVIL that consists of 4,678 pairs of facts, gist of claim, and the degrees of claim acceptance (Table 2). A model trained on this task can be used to assist judges or legal practitioners on this legal decision making process. The claim acceptance degrees are calculated (1) by parsing the claimed money from the gist of claim, (2) extracting the approved money from the ruling section, and (3) dividing the approved money by the claimed money. As a model requires diverse world knowledge for the accurate prediction of the numbers, we simplify this regression task by quantizing the claim acceptance degrees in two levels. At level 1, the claim acceptance degrees are partitioned into three categories: rejection, partial approval, and full approval (Table 7, 9th row). At level 2, the output ranges are further split into 13 categories with uniform range making the task more close to regression (10th row). To build the dataset, it is essential to extract moneys from the gist of claims and rulings which are not trivial to parse as they are often written in a compact form. For instance, instead of an easier expression such as "A provides $100 to C, and B provides $100 to C", judges would write "To C, A and B provides $100 each". To parse them we train mt5-small (Xue et al., 2021) with a prompt-tuning method (Liu et al., 2021; Ding et al., 2021) on 160 examples consisting of the pairs of gist of claim (or ruling) and the corresponding parses. The parses consist of the money provider, the money receiver, the amount of money, and the litigation cost. The ruling of criminal cases were parsed similarly. The related technical report is currently under preparation. We extract LJP-CIVIL from the precedents of following four categories: 929 examples from "price of indemnification" (구상금), 745 examples from "loan" (대여금), 1,004 examples from "unfair profits" (부당이득금), and 2,000 examples from "lawsuit for damages (etc)" (손해배상(기)). During the data collection, we do not collect precedents including counterclaims.

**SUMMARIZATION**    For legal practitioners, it is important to survey legal documents efficiently by focusing on the essential parts. In this regard, we make SUMMARIZATION dataset that consists of 20,000 pairs of precedent from the collection of Supreme Court Decisions Report containing Summary of Decision written by Director of Judicial Research from Supreme Court Library of Korea[12] (Table 2). In this task, a model needs to generate an abstract summary for the given ruling and the reasoning sections of the precedent. We retrieve the data from Korean Law Information Center[13] and collect precedents only when the sum of the number of tokens from the ruling and the reasoning section is less then 1,024[14]. The average number of tokens is 527 for the input texts (the ruling and the reasoning sections) and 133 for the ground truth (the summary). We also provide SUMMARIZATION+ by extending SUMMARIZATION with precedents with longer text making the task more challenging and realistic. In the extended dataset there are a total of 51,114 examples. The average number of tokens in the precedents and the corresponding summaries are 1,516 and 248 respectively. The maximum number of tokens in the input texts and the summaries are 93,420 and 6,536 respectively. The distribution of token length is shown in Fig. 2c in Appendix.

## 4    Experiments

### 4.1    Model training

All experiments are performed on Nvidia A6000, RTX3090 or RTX6000 hosted in Lambda, iwinv (https://www.iwinv.kr/), or nipa (https://www.nipa.kr/eng/index.do) clouds respectively. We use Transformers library Wolf et al. (2020) to fine-tune individual models. All models are trained with a constant learning rate 0.00003–0.0001 with batch size 8–16 using AdamW optimizer (Loshchilov and Hutter, 2017). All finetuning experiments with errorbars were repeated three time with different random seeds except LJP-CRIMINAL-Lv0 where the errorbars were estiamted from two independent experiments. To fine-tune mt5-small, `google/mt5-small` checkpoint is loaded. For LCUBE, we pre-train GPT-2 from scratch using Megatron library (Shoeybi et al., 2019) using PRECEDENT CORPUS together with Modu and Wiki corpora (Table 8). During pre-training the examples are randomly sampled from three corpora. We use morpheme-aware byte-level BPE for the tokenization reflecting that Korean is an agglutinative language (Kim et al., 2021a). We train LCUBE-base for 50K steps and LCUBE-medium for 100K steps which are well above the minimum

---

[12]https://www.scourt.go.kr/eng/supreme/decisions/guide.jsp
[13]https://www.law.go.kr
[14]The length is based on mt5 tokenizer (Xue et al., 2021)

Table 3: Comparison of various models. For the fair comparison, the test2 sets are used which are constructed from the precedents not included in PRECEDENT CORPUS. In case of SUMMARIZATION task, the summaries are not included in PRECEDENT CORPUS and thus the original test set is used. "imp.", "EM", "R1", "R2", and "RL" stand for imprisonment, an accuracy from exact match, Rouge-1, Rouge-2, and Rouge-Long scores respectively. "d.a." stands for "domain adaptation". "$n_{char}$" Stands for the average number of characters in the summary. The average $n_{char}$ of the GT is 242.

| Name | Size | CASE NAME (EM) | STATUTE (EM) | SUMMARIZATION | | | |
|---|---|---|---|---|---|---|---|
| | | | | R1* | R2* | RL* | $n_{char}$ |
| KoGPT-2-base | 125M | $78.5_{\pm 0.3}$ | $85.7_{\pm 0.8}$ | 47.2 | 39.1 | 45.7 | 277 |
| LCUBE-base-zero | 124M | $79.6_{\pm 0.6}$ | $85.8_{\pm 0.8}$ | 44.9 | 36.7 | 43.3 | 203 |
| mt5-small | 300M | $81.0_{\pm 1.3}$ | $87.2_{\pm 0.3}$ | 56.2 | 47.8 | 54.7 | 299 |
| mt5-large | 1.2B | $82.9_{\pm 0.2}$ | $88.1_{\pm 0.5}$ | 59.0 | 50.1 | 57.6 | 278 |
| KoGPT-2-base + d.a. | 125M | $81.9_{\pm 0.2}$ | $89.4_{\pm 0.5}$ | 49.2 | 40.9 | 47.7 | 273 |
| LCUBE-base-zero + d.a. | 124M | $80.4_{\pm 0.2}$ | $87.0_{\pm 1.7}$ | 46.2 | 37.6 | 44.6 | 200 |
| LCUBE-base | 124M | $81.1_{\pm 0.3}$ | $87.6_{\pm 0.5}$ | 46.0 | 37.7 | 44.5 | 195 |
| LCUBE-base + d.a. | 124M | $82.7_{\pm 0.6}$ | $89.3_{\pm 0.4}$ | 47.8 | 39.5 | 46.4 | 202 |
| LCUBE-medium | 354M | $81.2_{\pm 0.4}$ | $87.7_{\pm 0.5}$ | 50.1 | 41.4 | 48.6 | 213 |
| mt5-small + d.a. | 300M | $82.2_{\pm 0.2}$ | $88.8_{\pm 0.5}$ | 56.2 | 47.7 | 54.8 | 291 |

*: Computed at word level.

number of training steps expected from the power law (Ghorbani et al., 2022; Hoffmann et al., 2022). The model configuration is shown in Table 9. In the domain adaptation experiments, the KoGPT2 and LCube were trained with PRECEDENT CORPUS corpus for 14 epochs with the batch size 12 using the same objective (maximum likelihood with teacher-forcing). mt5-small was pre-trained with word level span corruption objective for 22 epochs with the batch size 12.

## 4.2 Task setting

We formulate the all tasks as text generation following (Raffel et al., 2020), not only because this allows an easy extension to other tasks but also it shows its effectiveness even in information extract tasks (Hwang et al., 2021; Kim et al., 2021b).

## 4.3 Metric

In CASE NAME, STATUTE, and LJP-CIVIL tasks, we use string exact match to evaluate the model accuracy. In LJP-CRIMINAL task, we calculate $F_1$ of individual fields (fine, imprisonment with labor, imprisonment without labor) as following. When the target field exists in both ground truth (GT) and prediction (1) the case is counted as a true positive if their values are equal and (2) false positive otherwise. (3) When the target field exists only in GT but not in the prediction, the case is counted as a false negative. (4) If the target field is empty in the GT but exists in the prediction it is counted as a false positive. (5) If the field is empty in both GT and the prediction, the case is considered as a true negative. The zero labels in LJP-CRIMINAL are counted as nulls.

## 5 Results

In this section, we show the baseline scores for CASE NAME, STATUTE, LJP-CRIMINAL, LJP-CIVIL, and SUMMARIZATION tasks by fine-tuning various language models (Table 3). We pre-train a decoder-only Transformer language model based on GPT-2 architecture. Decoder only models are known to show strong zero-shot performance compared to encoder-decoder language models and can be easily converted to non-causal decoder by additional pre-training for fine-tuning tasks (Scao et al., 2022; Wang et al., 2022). We prepare LCUBE-base, LCUBE-medium (Table 9), and LCube-base-zero, a twin model of LCube-base that is pre-traiend without using PRECEDENT CORPUS. We compare our models to KoGPT-2, a Korean GPT-2 made from SKT[15], mt5-small, and mt5-large (Xue et al., 2021) with or without domain adaptation.

**Domain specific corpus is critical in the classification and the summarization tasks** In CASE NAME classification task, LCUBE-base shows a superior performance compared to LCUBE-base-zero (+1.5%, 2nd vs 7th rows in Table 3) and KoGPT2 (+2.6%, 1st vs 7th rows). Similarly, in STATUTE classificatoin task, LCUBE-base shows +1.8% and +1.9% compared to LCUBE-base-zero

---

[15] https://github.com/SKT-AI/KoGPT-2

Table 4: Comparison of various models on LJP-CIVIL task. "EM" stands for exact match.

| Name | size | LJP-CIVIL Lv 1 (EM) | LJP-CIVIL Lv 2 (EM) |
|---|---|---|---|
| KoGPT-2-base | 125M | $66.0_{\pm 0.5}$ | $58.4_{\pm 0.1}$ |
| LCUBE-base-zero | 124M | $65.4_{\pm 0.9}$ | $58.6_{\pm 0.1}$ |
| mt5-small (512 gen) | 300M | $68.9_{\pm 0.8}$ | $57.7_{\pm 0.6}$ |
| mt5-large (512 g) | 1.2B | $69.0$ | $58.6$ |
| KoGPT-2-base + d.a. | 125M | $64.7_{\pm 1.1}$ | $55.2_{\pm 0.5}$ |
| LCUBE-base-zero + d.a. | 124M | $63.9_{\pm 1.5}$ | $52.7_{\pm 0.4}$ |
| LCUBE-base | 124M | $67.6_{\pm 1.3}$ | $58.8_{\pm 0.0}$ |
| LCUBE-base + d.a. | 124M | $60.9_{\pm 1.1}$ | $52.6_{\pm 0.4}$ |
| LCUBE-medium | 354M | $68.9_{\pm 1.6}$ | $58.2_{\pm 0.1}$ |
| LCUBE-medium + d.a. | 354M | $68.5_{\pm 1.2}$ | $55.7_{\pm 0.6}$ |
| mt5-small + d.a. | 300M | $69.1_{\pm 0.1}$ | $58.8_{\pm 0.2}$ |
| LCUBE-base - facts | 124M | $58.5$ | - |
| Most frequent label | - | $56.6$ | $57.3$ |

and KoGPT2 respectively. This clearly shows the importance and effectiveness of using domain specific corpus (PRECEDENT CORPUS) during the pre-training.

Next we perform domain adaptation experiments by further pre-training KoGPT2, LCUBE, and mt5-small with PRECEDENT CORPUS only. All models shows higher scores upon the domain adapation in (CASE NAME, STATUTE, SUMMARIZATION) tasks; KoGPT2 shows (+3.4%, +3.7%, +2.0 RL, 1st vs 5th rows); LCUBE-base-zero shows (+0.8%, +1.2%, +1.3 RL, 2nd vs 6th rows); LCUBE-base shows (+1.6%, +1.7%, +1.9 RL, 7th vs 8th rows); mt5-small (+1.2%, +1.6%, +0.1 RL, 3rd vs final rows). Notably, LCUBE-base shows a comparable performance with mt5-large, a competitive encoder-decoder language model with 860% more parameters (4th vs 8th rows).

Interestingly, in SUMMARIZATION task, LCUBE does not seem to have an advantage over other models (columns 5–7). Further analysis reveals, LCUBE has a tendency to generate ~40% fewer tokens compared to other models (2nd, 6–9th rows in the final column) which decreases the rouge scores. The performance of language models on downstream tasks is recently found to be dependent on pre-training corpus (Shin et al., 2022) which may account for the result[16]. The superior performance of mt5 in SUMMARIZATION task over other models may be originated from their architectural difference as encoder-decoder language models show stronger performance over decoder-only models on the tasks where understanding long context is necessary like a summarization task (Soltan et al., 2022). On CASE NAME+ task where the domain of the task extended by including infrequent case categories, LCUBE-base-zero achieves 75.6% exact match score (EM) whereas LCUBE-base achieves 78.9% EM. Similarly on STATUTE+ task, LCUBE-base-zero achieves 82.5% EM whereas LCUBE-base shows 84.9% EM. Again, both results show the importance and effectiveness of PRECEDENT CORPUS.

**Domain adaptation is not helpful on legal judgement prediction tasks** In LJP-CIVIL task, a model needs to predict the degrees of claim acceptance for the given facts and the gist of claim. The degrees of claim acceptance indicates the ratio between the approved money by the judge and the claimed money from the plaintiffs. To simplify the task, we first convert the task as a classification task having three labels: (1) label 0 for the case where the claim from plaintiffs is completely rejected, (2) label 1 when the claimed money is partially approved, and (3) label 2 when the claim is fully approved (9th row, Table 7 in Appendix). LCUBE achieves higher accuracy compared to LCUBE-base-zero (+2.2%) and KoGPT2 (+1.6%) (3rd column, 1st, 2nd rows vs 7th row). It is noticeable that the domain adaptation always decreases the accuracies of the decoder-only models on the contrary to the case of CASE NAME, STATUTE, and SUMMARIZATION tasks; -1.3% for KoGPT2 (1st vs 5th rows); -1.5% for LCUBE-base-zero (2nd vs 6th rows); -6.7% in LCUBE-base (7th vs 8th rows); -0.4% for LCUBE-medium (9th vs 10th rows). The accuracy of mt5-small also does not increase under the domain adaptation (3th vs 11th rows). This is in line with the result from the previous study showing that for more difficult tasks, pre-training from scratch is more helpful than domain adaptation (Chalkidis et al., 2020). When only the gist of the claims are used as the input to the model excluding the facts, the accuracy drops by 9.1% (7th vs 12th rows), highlighting the importance of the facts for the judgement prediction task. The test set consists of 56.6% rejection, 41.4% partial

---

[16]Although KoGPT-2 and LCUBE both were pre-trained using Modu and Korean Wikipedia corpora, KoGPT2 uses additional News, Petitions to the Blue House, and other unknown internal corpus.

Table 5: Comparison of various models on LJP-CRIMINAL task.

| Name | size | Lv 0 | | | Lv 1 | | | Lv 2 | | |
|---|---|---|---|---|---|---|---|---|---|---|
| | | fine | imp. w/ labor | imp. w/o labor | fine | imp. w/ labor | imp. w/o labor | fine | imp. w/ labor | imp. w/o labor |
| KoGPT-2-base | 125M | $69.0_{\pm0.7}$ | $83.2_{\pm0.3}$ | $82.6_{\pm0.2}$ | $61.0_{\pm0.3}$ | $72.7_{\pm0.0}$ | $59.3_{\pm1.0}$ | $49.9_{\pm1.7}$ | $67.5_{\pm1.1}$ | $69.2_{\pm1.6}$ |
| LCUBE-base-zero | 124M | $69.1_{\pm0.3}$ | $82.7_{\pm0.1}$ | $81.9_{\pm1.0}$ | $62.5_{\pm0.2}$ | $71.1_{\pm1.3}$ | $55.6_{\pm0.1}$ | 49.8 | 65.4 | 70.1 |
| mt5-small | 300M | $69.8_{\pm0.1}$ | $83.4_{\pm0.5}$ | $83.3_{\pm0.7}$ | $62.0_{\pm0.3}$ | $71.0_{\pm1.1}$ | $55.9_{\pm0.9}$ | $49.1_{\pm1.3}$ | $66.6_{\pm0.6}$ | $69.8_{\pm1.0}$ |
| mt5-large (512 g) | 1.2B | $69.6_{\pm0.3}$ | $83.5_{\pm0.6}$ | $84.0_{\pm0.5}$ | $61.7_{\pm0.3}$ | $72.4_{\pm0.5}$ | $58.1_{\pm0.4}$ | 46.9 | 68.7 | 69.5 |
| KoGPT-2-base + d.a. | 125M | $68.2_{\pm0.4}$ | $82.6_{\pm0.8}$ | $83.0_{\pm0.4}$ | $60.9_{\pm0.1}$ | $71.5_{\pm0.0}$ | $55.6_{\pm0.6}$ | $51.1_{\pm0.9}$ | $65.7_{\pm2.5}$ | $68.8_{\pm1.0}$ |
| LCUBE-base | 124M | $67.2_{\pm2.6}$ | $82.2_{\pm0.6}$ | $81.5_{\pm0.2}$ | $60.1_{\pm1.7}$ | $71.0_{\pm0.3}$ | $55.2_{\pm0.3}$ | $46.4_{\pm2.8}$ | $69.3_{\pm0.3}$ | $70.3_{\pm0.7}$ |
| LCUBE-base + d.a. | 124M | $66.8_{\pm1.7}$ | $80.7_{\pm0.4}$ | $81.6_{\pm1.2}$ | $59.4_{\pm2.1}$ | $69.3_{\pm0.4}$ | $55.9_{\pm3.5}$ | $48.1_{\pm1.2}$ | $67.4_{\pm1.5}$ | $69.9_{\pm1.1}$ |
| LCUBE-medium | 354M | $69.2_{\pm1.9}$ | $81.9_{\pm1.5}$ | $82.7_{\pm0.3}$ | $61.2_{\pm2.1}$ | $70.8_{\pm1.8}$ | $57.6_{\pm0.7}$ | $49.3_{\pm0.8}$ | $67.3_{\pm1.9}$ | $68.1_{\pm1.0}$ |
| LCUBE-medium + d.a. | 354M | $69.6_{\pm1.4}$ | $83.8_{\pm0.1}$ | $84.0_{\pm1.1}$ | $61.4_{\pm0.4}$ | $72.8_{\pm0.6}$ | $53.2_{\pm0.3}$ | $48.6_{\pm2.0}$ | $69.3_{\pm0.1}$ | $69.8_{\pm0.5}$ |
| mt5-small + d.a. | 300M | $67.9_{\pm0.4}$ | $82.1_{\pm1.1}$ | $80.8_{\pm0.6}$ | $61.3_{\pm1.1}$ | $71.2_{\pm1.1}$ | $54.0_{\pm3.3}$ | $51.8_{\pm0.7}$ | $68.9_{\pm0.3}$ | $70.3_{\pm0.1}$ |
| Most frequent label | - | 56.6 | 67.9 | 15.5 | 34.3 | 45.2 | 8.8 | 30.1 | 42.8 | 11.9 |
| LCUBE-base - facts + casename | 124M | 63.5 | 75.0 | 74.6 | 55.3 | 65.6 | 49.4 | 42.0 | 64.0 | 61.4 |
| LCUBE-base + reason † | 124M | 77.5 | 87.2 | 78.7 | 69.1 | 76.2 | 53.0 | 60.5 | 74.6 | 72.7 |
| LCUBE-medium + reason † | 124M | 81.8 | 87.5 | 86.6 | 74.0 | 77.3 | 63.1 | 60.4 | 76.1 | 76.1 |
| Null ratio | - | 0.605 | 0.486 | 0.916 | 0.605 | 0.486 | 0.916 | 0.605 | 0.486 | 0.916 |
| Top non-null label ratio | - | 0.395 | 0.514 | 0.084 | 0.207 | 0.292 | 0.046 | 0.188 | 0.272 | 0.063 |
| Top non-null label ratio (w/o null) | - | 1.0 | 1.0 | 1.0 | 0.523 | 0.568 | 0.551 | 0.474 | 0.528 | 0.744 |

† The reason for the sentencing.

accept, 2.0% full accept. This indicates that without the facts, the model performance is close to a dummy baseline that selects the most frequent label (i.e. rejecting every claim).

To check whether the result is not from an artifact of the quantization of the claim acceptance ratio, we perform additional experiments with more fine-grained labels (Table 7 in Appendix, final row). Under this new setting, the claim acceptance ratio is partitioned with 0.1 interval mimicking the regression setting (when the range is quantized uniformly and their interval becomes infinitesimal, a classification task becomes a regression task (Bishop, 2006)). Again we found the domain adaptation is not helpful; -3.2% for KoGPT2 (1st vs 5th rows, final column); -4.1% for LCUBE-base-zero (2nd vs 6th rows); -6.2% in LCUBE-base (7th vs 8th rows); -2.5% for LCUBE-medium (9th vs 10th rows); +0.2% for mt5-small (4th vs 11th column). Similarly, on LJP-CRIMINAL task, the domain adaptation does not show clear improvement on the model performance (see next section).

**Legal judgement prediction is challenging**  In LJP-CRIMINAL task, a model needs to predict the fine amount, the range of imprisonment with labor or imprisonment without labor for the given facts. Given the difficulty of the task, we approach the task with three different levels of difficulty: (1) level 0 where a model only needs to predict the type of punishment if any, (2) level 1 where a model predicts the punishment in three degrees (null, low, high), and (3) level 2 where the ranges of punishments are further partitioned into five (fine) and six (imprisonment) labels. Not surprisingly, the scores decrease at higher level (Table 5). Except at level 0, the models show much lower performance compared to CASE NAME and STATUTE. To confirm the importance of the facts for the prediction, we replace the facts in the inputs to the case names. The $F_1$ scores for fine, imprisonment with labor, and imprisonment without labor prediction tasks drop by -3.7%, -7.2%, and -6.9% at the level 0, -4.8%, -5.4%, and -5.8% at the level 1, and -4.4%, -5.3%, and -8.9% at the level 2 (6th vs 12th rows). This clearly indicates that the facts include importance information for the judgement prediction. When the predicted labels for the individual subtasks are replaced by the most frequent non-null labels for all cases, the $F_1$ scores decrease further (11th row). The relatively higher scores observed when the case name are used as the inputs may be originated from the fact that the cases with the same category will likely to have relatively similar punishments ranges. As another control experiment, we add "the reason for the sentencing" section of the precedents to the inputs. In this section, the judges describe various aspects that can affect the final results such as upper and lower bounds of punishment ranges described in the related statutes, ages and attitude of defendants, victim's opinions etc. This can be considered as an incomplete oracle. Upon the addition of the oracle, there is dramatic increase in $F_1$'s (6th vs 13th rows, and 8th vs 14th rows). Interestingly, unlike other tasks, we could not find the single superior model even with additional experiments under regression setting (Table 10 in Appendix).

## 6    Conclusion

With the rapid advancement of deep learning, how artificial intelligence (AI) can be applied in legal domain is fundamentally changing. In response to this, we release the first large-scale Korean legal

AI benchmark LBox Open and Korean legal language model LCube. LBox Open consists of one legal corpus, two text classification tasks, two challenging legal judgement predictions tasks, and one precedent summarization task. Experiments on various language models highlight the importance of pre-training language models on domain-specific corpus in large scale. In the future, we plan to extend LBox Open to include more diverse legal language understanding tasks such as court opinion generation.

## Limitation and Future Works

In this work, we only consider the precedents from the first level courts. In the trials from the second level courts, the legal reasoning can become more complex due to the combination of different appeals from each party. We also do not use claims from plaintiffs and defendants as an additional input to a model although they include important information for various legal decisions. Without a high-performance parser, it is difficult to separate the claims from reasoning sections without errors as they are loosely separated from the facts and the reasoning. The claims will be integrated to the future version of LBox Open. Our datasets also currently do not consider many important legal applications of AI such as a legal information retrieval task. This will be integrated in LBox Open in the future.

## Ethical Considerations

The precedents from criminal courts often include detailed descriptions about crimes. Thus one can obtain the essential information for the illegal acts by reading the related precedents. Nevertheless, laws alone cannot comprehensively cover complex social phenomena and precedents often fill the empty spaces of the laws. Thus, we believe, enabling an easy access to precedents can bring more social benefits than harms.

The models trained on the legal judgement prediction (LJP) datasets can be used for automatic legal decisions and like any data-driven approaches, the models can predict biased results depending on the data statistics of the training set. To minimize the risk, we have used only the precedents redacted by the Korean government following the official protocol (Section 2.2). Also, we have fully open-sourced the dataset, trained model, and training code so that anyone can access and investigate the dataset and the method without any restriction.

Nevertheless as the population of precedents is already biased reflecting the current status of the society (for example, the judicial yearbook published in 2021 reveals the 86.8% defendants in 1st criminal trials were male[17], and as we randomly sampled the population during the construction of LBox Open whenever possible, the use of the released dataset and the model without explicitly understanding the potential issue from the dataset bias can make negative social impacts.

To concretely address this issue, we performed a case study about gender bias in the criminal precedents (race, language, and regions of litigants are not considered as our datasets consist of Korean cases only). The age is also not considered at this time as judges can reduce or increase the sentences based on the ages of litigants following the official sentencing guidelines.

We first extracted the victims' gender information from the facts using regular expressions. Among five criminal case categories in LJP-CRIMINAL dataset, only "indecent act by compulsion" (강제추행) shows strong bias in victims' gender (1,451 females, 74 males). We examined how the prediction of LCube changes when the victims' genders are exchanged (as only 0.46% cases include the gender of the defendant in the facts, we change the gender of the victims). Among 131 test examples, the prediction changed in 11 examples. Further analysis revealed in 10 examples (7.6%) the outcome sentences decreased when the victims' genders were changed from female to male. Although this is relatively small (91.5% of the predictions remain identical), the result shows the potential risk and the limitation of automatic legal decision making based on our dataset and the released model. Finally we would like to remark that there is also a social benefit in democratizing the data.

---

[17]https://www.scourt.go.kr/img/pub/jur_2021_Book6.pdf

## Acknowledgments and Disclosure of Funding

We thank Gene Lee for his helpful comments and advisory (as a licensed lawyer in Korea) on the legal aspects of this work and carefully proofreading the paper. We also thank the anonymous reviewers for their critical comments.

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
