# OpenReview forum: "A Multi-Task Benchmark for Korean Legal Language Understanding and Judgement Prediction"
_NeurIPS.cc/2022/Track/Datasets_and_Benchmarks — NeurIPS 2022 Datasets and Benchmarks _

### Official Review · Reviewer_xaXW · 2022-07-04
**Review - A Multi-Task Benchmark for Korean Legal Language Understanding and Judgement Prediction**

**Rating:** 9
**Confidence:** 4
**Correctness:** The evaluation methods and experiment…
**Clarity:** Yes.

**Strengths:**

The paper contribution consists of the introduction of a large-scale corpora, benchmark datasets, and language models in the domain of Korean Law. Legal datasets are scarce, particularly in different languages than the English language. This dataset offers an important resource for empirical legal researchers and computational researchers who wish to analyze Korean case law. The six datasets that are introduced (and benchmarked) have the potential to provide for a significant breakthrough. The same applies to the Korean legal language model that is released. To my knowledge, it is the first of its sort. It was quite the endeavor to compose the datasets and the model, which makes the construction a significant effort that is unlikely to be done by others.

One could argue that a Korean legal dataset has limited relevance to the broader research community. By that standard, the vast majority of non-English datasets can be excluded. As the authors point out, there is a knowledge gap with respect to NLP models (and data) in languages that are not English.

The dataset is made available on Github. Although I have not ran the code myself, the documentation seems sufficiently clear and transparent in order to access the datasets.

**Weaknesses:**

The authors speak about precedents, but it does not become entirely clear what is meant with precedents. As the authors rightfully point out, precedents are cases from the past that have produced a rule that can be used to decide subsequent cases. Many cases are not considered precedents, although on a generic level every court decision can be seen as a precedent. It seems that the author follow the latter approach, essentially equating precedents with court cases. This is fine, but it would be good if the authors would explain this. This is a minor comment.

The paper speaks about 6k precedents that were collected as part of approximately 1.5 million Korean precedents in the total population. Maybe I missed it, but I did not encounter possible explanations for the discrepancy. For instance, could selection bias be involved? It remains difficult to determine the generalizability of the dataset.

I am not sure what is measured in the 'prediction' part. What is predicted, is the outcome of the decision based on the text that the court produced. This is more likely to be a prediction of how judges motivate their decisions in case of a certain outcome rather than predicting outcomes based on 'the facts'. The results in the 'LCUBE-base + reason (oracle)' model are surprising. As previous research shows, one would expect that the outcomes of court cases can be predicted with high levels of accuracy when including the reasoning of judges. Or did I not correctly understand what was included in the 'reason (oracle)' here?



**Additional Feedback:**

The paper contains some typo's.

**Documentation:**

Documentation is sound. The dataset can be accessed.

**Ethics:**

No ethical concerns.

**Relation To Prior Work:**

The paper would benefit from including more references to studies that aim to predict outcomes of court decisions.

**Summary And Contributions:**

Large-scale corpora, benchmark datasets, and language models in the legal domain are lacking. This paper remedies this by introducing this for Korean Law. It introduces Korean legal AI benchmark datasets that consist of six datasets: one large-scale legal precedent corpus, two classification tasks, two legal judgement prediction tasks, and one summarization task. In addition, the a large-scale Korean legal language model is released. As part of the (pre)processing, the authors processed document images and PDFs. The pipeline is presented and explained.

---

> ### Author Response · Authors · 2022-08-24
> **Respone to the reviewer xaXW**
>
> We are grateful to the reviewer for recognizing the value of our work.
>
> [About the meaning of "precedent" in Korean legal system]\
> As the Korean legal system is rooted in civil law system, the court decisions on previous cases only play supplementary roles and are not institutionally adopted in Korea. However, the official Korean-English translation on legal terms use "precedent" to indicate the previous cases and thus we adapted "precedent"(https://www.moleg.go.kr/board.es?mid=a10504000000&bid=0010&act=view&list_no=43927&nPage=2). In Korean, it is called "판례(判例)" and their literal meaning is "example of decision".
>
> [(3-2) Precedent Disclosure Status]\
> According to Civil and Criminal Procedure Act, the courts are making their judgement decisions public via online service (https://www.scourt.go.kr/portal/information/finalruling/guide/index.html) provided by the Supreme Court of Korea. Despite this, the accessibility of the Korean precedents to the general public is still contentious in that the courts only provide final decisions that are irrevocable while charging a 1,000 won (\~1 USD) fee per a precedent. To have access to full contents of 100k precedents, one needs to pay approximately \~100k USD (incidentally, this indicates \~70k precedents we have released cost 70,000,000 won just for the puchasement of the raw data). Thus, although 1.5m Korean precedents are created every year, it is challenging to get access to the full data with a  limited amount of budget. Nevertheless, the dataset was constructed via random sampling reflecting the real data distribution as much as possible.
>
> [What is measured in the prediction part]\
> In Korean precedents from 1st criminal trials, there is a section titled "the reason for sentencing". Here the judge describes the range of possible punishment for a given crime, the reason for increasing/decreasing sentencing period without explicitly mentioning exact imprisonment period or fine amount. However, the official sentencing guideline (https://sc.scourt.go.kr/sc/krsc/criterion/criterion_01/murder_01.jsp) only provides the lower and upper bounds of possible punishment ranges divided by three high levels (reduced, base, weighted). Thus, it is possible that even with given facts and additional reasoning, there can still be differences in sentencing between judges. See also https://scholarship.law.ufl.edu/flr/vol69/iss1/2/ that discusses implicit bias in judicial decision-making.

---

### Official Review · Reviewer_vj1q · 2022-07-06
**interesting benchmark with solid contribution on building AI systems around legal services**

**Rating:** 6
**Confidence:** 4
**Correctness:** The claims made in the submission see…
**Clarity:** This paper is well written.

**Strengths:**

- The main contribution is that the authors contribute a large open-source LBOX OPEN dataset in Korean for evaluating language understanding for legal documents.
- This is valuable given that legal documents are hard to harvest and annotate by humans.
- It also contributes to multi-lingual studies over language understanding for legal documents.

**Weaknesses:**

- Baselines still suggest that a multi-lingual model such as MT5 performs better than in-distribution pretrained LCUBE-base. It is worth exploring even large models such as GPT-2 or GPT-3, if not MT5-base or large, to benchmark existing multi-lingual models with this benchmark.

- It would be interesting to see how an English pre-trained language model performs by directly asking it to transfer to do legal documents understanding in another language (assuming we have some hacks in tokenization).

- It would be interesting to see more error analyses over these baselines. For instance, what are the cases that MT5 gets wrong in terms of CASE NAME classification? Is this because the document is too long? Or it contain complex descriptions in Korean?



**Additional Feedback:**

No.

**Documentation:**

I found sufficient detail on data collection and organization, availability and maintenance, and ethical and responsible use.

**Ethics:**

There are no ethical concerns to me.

**Relation To Prior Work:**

Related works are well discussed.

**Summary And Contributions:**

The authors present a multi-task legal service benchmark in Korean for language understanding measures over judgment prediction and many other tasks. The authors also build baselines using existing mono-lingual models as well as multi-lingual models and show enough headroom for future research. The main contribution is that the authors contribute a large open-source LBOX OPEN dataset in Korean for evaluating language understanding for legal documents. This is valuable given that legal documents are hard to harvest and annotate by humans. It also contributes to multi-lingual studies over language understanding for legal documents.

---

> ### Author Response · Authors · 2022-08-24
> **Response to the reviewer vj1q**
>
> We thank for the critical comments. We have added new experimental results and analysis with new models  LCube-medium (124M -> 345M), mt5-Large, and mt5-small with domain adaptation. Below we present our replies to individual comments.
>
> [The additional scaling and domain adaptation experiments]\
> Please refer to our replies above.\
> [(1-1) Additional scaling and domain adaptation experiments with other models]\
> [(1-3) More experiments and analysis on the summarization task]\
> [(1-2), (3-3) The effects of label quantization on the LJP tasks]\
>
>
>
>
> [Error analysis on the case name classification task]\
> On the case name classification task, LCube-base achieves 81.1% overall accuracy whereas mt5-small achieves 81.0% acc. However, the accuracy of individual case name are different from each other. For instance, on "aiding fraud(사기 방조)" class, the accuracy of individual models are as follows.
> \
> —---------------------\
> kogpt2: 16.7% (overall 78.5%)\
> LCube-base w/o legal corpus: 83.3% (overall 79.6%)\
> mt5-small: 25.0% (overall 81.0%)\
> —----------------------\
> LCube base: 83.3% (overall 81.1%)\
> kogpt2 + domain-adaptation: 100% (overall 81.9%)\
> mt5-small + domain-adaptation: 58.3% (overall 82.2%)
>
> The results indicate that the use of legal corpus greatly increases the accuracies in kogpt2 and mt5-small models whereas LCube-base already archives high performance without domain adaptation. This indicates the performance of language models are affected not only by domain corpus, and model architecture but also by the other copus used during the pre-training as recently found in "Shin et al., On the Effect of Pretraining Corpora on In-context Learning by a Large-scale Language Model, 2022".
>
> [Cross-lingual experiments]\
> Thanks for the interesting suggestions. However, the size of  current language models (<1B) are generally reported to show weak zero-shot or in context learning performance. Once we are able to scale LM up to ~10B, we'll investigate this interesting topic.

---

### Official Review · Reviewer_7fpy · 2022-07-14
**LBOX - An interesting proposition but lacking rigor**

**Rating:** 4
**Confidence:** 4

**Strengths:**

The corpus released as well as the processing into the various tasks is a new addition and composed of marginally more data than the previous dataset for Korean law from AI-hub, ranging from 11K to 150K data points. The new language model for Korean legal language is also of practical interest.

**Weaknesses:**

The whole pipeline, from the preprocessing to the evaluation, lacks rigors and makes me doubt the pertinence of the contribution. More specifically:
- The extraction from raw data is produced by ML, with an interconnection of systems and some manually labelled pages. There is no mention of how well each of these systems perform, and no verification on how well they perform together (for example, verification on a 100 randomly selected examples that the extraction was correct). It would also be good to have information such as the threshold for manual labelling, as well as the reasoning behind the choice of that particular threshold.
- For the tasks, only a limited range of examples are included ("100 most frequent case categories", "46 frequent case categories", etc). No justification for these choices are provided; these limitations are not needed since all tasks are modelled as a language modelling task which can be trained with any number of categories; the limitations diminishes greatly the interest for the dataset outside academic research; the task doesn't reflect the performance it would have would it be used in practice.
- The language model LCUBE-base is evaluated and compare to KoGPT2-base on the new tasks, which are part of the corpus used to train LCUBE-base. Given this, it would be surprising if LCUBE wouldn't outperform KoGPT2-base. While limited, an already better evaluation would be to evaluate on the dataset of AI-hub which should be a legal task but from a different corpus.
- One of the argument for this paper is that the dataset from AI-hub is limited in data given that "1.5 million Korean precedents are generated per year". However, the dataset provided is composed of 150K precedents, which doesn't even amount to one year of Korean precedent. The question naturally arises as to why not have a bigger corpus if the data is here and manual labelling is limited in the creation process?
- There is no analysis provided which would help underline the performance of the system and how it could be used.

**Additional Feedback:**

At the minimum, the work to be done should be
- A solid rereading for errors
- A better justification of the limitation for the task
- More transparency in the creation process
- Better evaluation of the language model
- Better framing of the ethical problem
- A related work section that justifies more the various tasks

Optimally, the tasks would be re-run without the limitations and with more analysis provided, for example on how the system perform on low-frequency classes.

**Clarity:**

Correction are needed as mistakes are often found (ex "We pre-trained GPT-2 from the scratch"), the abstract and introduction are dense and not very clear, and rewriting them would be beneficial, but the rest of the paper is clear.

**Correctness:**

As stated above, several choices in the construction of the dataset are dubious and, if not changed, need strong justifications. Furthermore there are several claims in the paper that are made that should have a citation as they are not common knowledge, ex: "Korea has one of the largest legal industries in the world", or "In the real world, automatic categorization of legal documents or legal questions are often important for various downstream tasks.".

**Documentation:**

There is a concerning lack of details for the collection process, relating to the ML process used. Datasheet is present and complete but given that one of the argument for this corpus is that massive amount of data are released each year and that "it is difficult to expect the released precedents can cover various legal activities", it should be discussed in more details if the dataset is going to be updated, lest it becomes obsolete in a few years.

**Ethics:**

The ethical section significantly downplays the possible negative impacts of legal ML (ex https://www.technologyreview.com/2019/01/21/137783/algorithms-criminal-justice-ai/ ). Given the overall known problems of discrimination in Korea the paper should at least warn that this dataset shouldn't be used without serious fairness assessment so it doesn't amplify and reproduce biases.

**Relation To Prior Work:**

It is clearly discussed how the work differs from the corpus of AI-hub, but the paper would benefit from a section discussing Korean legal ML or even larger legal ML in order to better establish the relevance of the various tasks.

**Summary And Contributions:**

The authors propose a new legal corpus for the Korean language, with a total of 5 associated tasks: two classification tasks, two legal judgment prediction, and one summarization task. They furthermore release a new language model based on GPT-2 architecture but for Korean legal language.

---

> ### Author Response · Authors · 2022-08-24
> **Response to the reviewer 7fpy**
>
> We thank the reviewer for the careful reading of the paper and several suggestions.
> Below we provide our replies to the major comments including (1) how we have extended the domain of two classification datasets, (2) the practical limitation in getting access to the raw data in scale, and (3) additional description of the ML modules in our data engineering pipeline.
>
> \
> [(2-3), (2-4) Extending the range of examples in case name and statutes classification tasks]\
> In our initial study, we only included a limited range of case categories (100 in the case name classification, and 46 in the statute classification) to set the task in a well-controlled environment to avoid dataset bias from the data imbalance. Upon feedback, we have extended two datasets by including infrequent case categories making the task more realistic.
>
> This results in 100 -> 603 case categories, 8,000 -> 22,494 training examples, 1000 -> 3,999 validation examples, 1000 (test), 1294(test2) -> 4,790 test examples in the case name dataset. The top two most frequent categories are fraud (1,885), lawsuits for damages (etc) (1664). On this new dataset LCube without legal corpus achieves 75.6% acc, LCube-base 78.9% acc, LCube-medium 77.7% acc, mt5-small with domain adaptation 81.0% acc confirming again, the effectiveness of the precedent corpus.
>
> In the new statute dataset, we have 2,208 -> 13,317 training examples, 266 -> 2,276 validation examples, 276 (test), 538 (test2) -> 2,137 test examples. The top two most frequent categories are fraud (1,686) and drunk driving (1,442). LCube without legal corpus achieves 82.5% acc, LCube-base 84.9% acc, mt5-small with domain adaptation 82.0% acc.
> \
> [Precedent disclosure status]\
> There are practical difficulties in accessing the precedents in scale.
> Please refer to our reply "[(3-2) Precedent Disclosure Status]" below.
>
>
>
> [(2-5)The detail of the data engineering pipeline]\
> Here we provide the additional detail of the precedent engineering pipeline.
> Layout-classifier is prepared by training custom ResNet using 300k training examples consisting of document image and label pairs.  For Layout-parser, we train Mask-RCNN (Detectron library from facebook was used) on 162k examples consisting of document image and coordinates of the figures/tables pairs. For Post-OCR Text corrector, we use character level transformers trained on 137 k examples that consist of OCR outputs and manually corrected text pairs. The transformer generates tags indicating whether to "keep", "delete", "replace", or "insert" for individual characters. When the generated tags are either "replace" or "insert", the transformer generates an additional new character. All training sets were prepared via our own data labeling platform LWorks (https://lworks.kr/work) where we employ ~100 part-time annotators.
>
> The precision and recalls of individual ML modules on the static test sets are as follows.
> Layout-classifiers, P 99.94%, R 82.0%
> Layout-parser, P98.6%, R97.1% with IOU threshold 0.5
> Post OCR Corrector, P 91.1%, R74.6%
>
> The overall automation efficiency on the deployed system is ~80%  for PDF-type precedents and ~53% for non-readable type precedents. The remaining precedents are manually monitored and corrected via the LWorks platform.
> We are currently preparing a separate technical report explaining the details of the method.
> \
>
> [Additional experiments and analysis]\
> Please refer to our following two replies from above.
> "[(1-3) More experiments and analysis on the summarization task]"
> "[(1-2), (3-3) The effects of label quantization on the LJP tasks]"
> \
>
> [The market size of Korean legal industry]\
> By 2020, the Korean legal industry market size is estimated (based on the value-added tax base declaration amount) as ~$4.8B (https://m.lawtimes.co.kr/Content/Article?serial=163743) whereas Japan ~$3.5B in 2020
> (https://www.researchandmarkets.com/reports/5503203/legal-services-in-japan-market-summary), Singapore ~$1.7B (2019), India ~$1.3B (https://en.wikipedia.org/wiki/Legal_industry_by_country#India). Unfortunately, we could not find official documents reporting the market size of Asian legal industry. We'll change the sentence from "Korea has one of the largest legal industries in the world" to "Korea has ~$4.8B industrial market size".
> \
> [Data maintenance plan]\
> We initially released LBox Open on Mar. 2022 with one corpus and three tasks and have updated the dataset by adding two LJP tasks and LCube LM on Jun. 2022. The newer data is expected to be released with 0.5–1 year interval as our internal dataset and the engineering quality grow.
> \
>
> [Ethical section]\
> Thanks for the critical comments and the reference. Please refer to our reply to the ethics reviewer oMd1.

---

### Official Review · Reviewer_2huR · 2022-07-24
**A nice job in the direction of Korean legal Benchmark**

**Rating:** 6
**Confidence:** 3
**Correctness:** basically correct.

**Strengths:**

1. The dataset covers enough case categories, and the data structure is clear and easy to use.
2. The proposed tasks have real business value and solve real business problems.
3. This work has a certain role in promoting the application of AI technology in the legal direction.

**Weaknesses:**

1. I feel a little weak in the experimental part. It is recommended to add a pre-training of the encoder-decoder language model for comparison.
2. On the summary task, the reason why LCUBE is significantly lower than other models can be further analyzed, or some other evaluation metrics of the generation tasks or even some artificial evaluation metrics can be introduced.

**Additional Feedback:**

Good datasets and tasks, it would be better if the experimental part can be enriched, such as using more structured models, trying large models, or analyzing the phenomena shown in the experimental results in more depth.

**Clarity:**

There are some minor flaws, such as a spelling error in the second line of Section 4.1(fine-tnue), and a spelling error in the Fine-tuning section of github's README(pytyon).

**Documentation:**

The dataset has clear enough relevant documentation and usage instructions, and the hyper-parameters and codes used in each task are also open source.

**Ethics:**

In the last part of the article, the author made some ethical considerations and expositions. I basically agree with the author's point of view, enabling an easy access to precedents can bring more social benefits than harms.

**Relation To Prior Work:**

Yes, in Section 3.2.

**Summary And Contributions:**

This work proposes a dataset (LBox-Open) containing 150k Korean legal precedents, and proposes two classification tasks, two legal judgment prediction tasks, and a summary generation task based on part of the data in this dataset. And a GPT2 model LCUBE-base is pre-trained using three datasets including LBox-Open. Both the model and the data are open source and can be easily loaded and used through HuggingFace.

**Datasets**
Their proposed dataset is the first large-scale Korean-language legal-orientation dataset that combines scale and diversity. Contains 63k cases sentenced in the past four years and 96k cases from the first and second level courts in the fact-checking stage, of which a total of 80k are from LAW OPEN DATA searches and 70k are from their own database, covering a total of 1,160 case categories. They have customized a set of data engines to normalize the data format, and uniformly process the data in the original format of pdf and pictures into the format of text.

**Tasks**
Based on a 150k formatted dataset, they propose 5 tasks. All are based on actual business, and four of them are classification tasks, (1). Predict the type of cases based on facts, (2). Predict violations of laws  based on facts, (3). Predict fines, prison months, and labor imprisonment months based on facts. (4) Predict claims acceptance based on facts and gist of claim. One is the generation task, which is to predict summaries based on ruling and reasoning section.

**Model**
Based on the three datasets of Wiki, Modu and LBox-Open, they pre-trained a GPT-2 with 124M parameters and compared it with KoGPT-2-base and MT5-small on the above five tasks, providing a task baseline.

---

> ### Author Response · Authors · 2022-08-24
> **Response to the reviewer 2huR**
>
> We thank the reviewer for recognizing the value of our dataset in task formulation in the real world. Here we provide our answer to the major comments.
>
>
> [(1-1) Additional scaling and domain adaptation experiments with other models]\
> We performed additional scaling (LCube-base -> LCube-medium) and domain adaptation experiments with KoGPT2 and mt5-small. Here, the domain adaptation indicates the pre-training with the Precedent corpus starting from the already pre-trained model on the general domain. We found the domain adaptation increases the performance in the case name, statute classifications, and the summarization tasks across all models. This proves the usefulness of our legal corpus. We also find the scaling (LCube 124M -> 354M) leads to similar performance in the two classification tasks but better rudge scores in the summarization task (+2.3 R1, +1.9 R2, +2.2 RL). In the case name classification task, mt5-large (1.2B) and LCube-base with domain adaptation show best performance whereas in the statute classification tasks, KoGPT2 +domain adaptation and LCube-base outstand. In the summarization task, mt5-large and mt5-small with domain adaptation shows the top performance.
>
>
> For two LJP tasks, please refer to "[(1-2), (3-3) The effects of label quantization on the LJP tasks]" above. The full experimental results will be available from the next revision.
>
>
> [More experiments and analysis on the summarization task]\
> Please refer to our reply "[(1-3) More experiments and analysis on the summarization task]" above.

---

### Official Review · Reviewer_9YQQ · 2022-07-25
**A useful resource, good documentation, with some missing points**

**Rating:** 7
**Confidence:** 4

**Strengths:**

First such large-scale domain-specific benchmark in Korean with good documentation. Existing legal tasks are largely English-oriented. Having legal NLP tasks in a diverse set of languages is important.

**Weaknesses:**

* Apart from the specificity of the language, the motivation for this new benchmark is not explicit. The authors propose several legal tasks based on the data they were able to collect but do not emphasize the importance of these tasks in the real world and how they could benefit society (apart from the summarization task, perhaps).

* As the authors state in Section 3.1, structuring raw data that contains a mix of plain text, tables, and images is a non-trivial task and represents an important part of the work and value provided in the paper. However, little detail is given regarding the different components of the data engineering pipeline (no information about the specific checkpoints or model parameters used for the layout classifier and parser, the OCR framework, and the model used to correct OCR errors). Additionally, code for extracting quality data from the raw documents does not seem publicly released (unlike code for experiments), which is a pity as it would provide excellent value for the research community.

* Authors claim that LCube is a legal language model. However, it turns out that the legal corpora only represent 5.5% of the whole pre-training corpus, whereas news and books represent 90% of the pre-training data. I wonder why the authors did not start from a pre-trained KoGPT-2 checkpoint and further pre-trained on their legal corpus only, which has been shown to lead to similar or even better results than pre-training from scratch for a portion of the computing resources — see ClinicalBERT (Huang et al., 2019), BioBERT (Lee et al., 2020), SciBERT (Beltagy et al., 2019).

Other nits:

* In the Abstract and Introduction, the authors mention some approximative dataset sizes that exceed the actual sizes of the datasets (e.g., 3k pairs for Statute while 8% smaller in reality, 11k pairs for LJP-Criminal while 4.5% smaller in reality, 5k pairs for LJP-Civil while 6.4% smaller in reality). Authors should prefix these approximations with "around"/"~" in order not to mislead the reader about the actual sizes of the provided datasets.

* For LJP-Criminal, the authors state that unavailable data (such as the age of defendants) makes the regression task too difficult and therefore formulate it as a classification task by discretizing the outputs. However, there does not seem to have been a preliminary investigation that supports these claims. Also, the authors explain that the boundaries for quantization are chosen in part based on Korean legal aspects and give the example of 1M won as the lower bound amount where public officials can lose their position if found guilty. This would be interesting to not only have one example of such legal aspects but a clear description of all of them for each bound they correspond to. Additionally, authors should show the data distribution related to those boundaries (to make sure that the models do not simply learn to predict the over-represented classes). Same remark for LJP-Civil.

* The Results section feels like it was written relatively quickly: it lacks detailed analysis and is poorly organized. The authors performed good control experiments that deserve their own subsection with a deeper analysis of the results. Similarly, a more detailed comparison between KoGPT-2 and LCube is missing.

**Additional Feedback:**

* The Korean legal system explanation (Section 2.1) lacks clarity about the difference between civil and common law and the concept of stare decisis. The reader (who is expected to have very little legal knowledge) might benefit from a more detailed explanation of those topics. Same remark for the "no penalty without a law" principle, mentioned at page 4.
* "In the real world, automatic categorization of legal documents or questions are often important for various downstream tasks" (p.4). Authors should give a few concrete examples of such important downstream tasks.
* References to "Appendix B.x" actually correspond to "Appendix A.1.x" in the paper.
* Regarding the statute prediction task, it is mentioned (p.4) that the dataset is built from the same precedents used in CaseName. However, the Statute dataset only contains samples from 46 frequent case categories with 60 examples each, unlike the 100 most frequent case categories with 100 examples each from CaseName. The authors do not explain the choice of reducing both the number of case categories and examples per category for that second text classification task. Also, I assume that the 169 classes stated in Table 3 correspond to the number of different statutes found in those 46*60=2760 samples. If so, it would not be wrong to mention it for clarity.
* For LJP-Civil, the authors explain that they performed the extraction of the amounts of money by "preparing their custom language model trained on a few hundreds of examples of ruling/claim-money pairs. The sentence is very unclear. What do you mean by "preparing", did you fine-tuned LCube on a regression task using these samples?
* For Summarization, the authors do not mention how the summaries were generated. Did the legal experts from the Korean Law Information Center write them? Also, it is a pity that the authors did limit their collection to precedents whose ruling and reasoning sections were less than 1024 tokens, their choice being based on the maximum input length of a specific model at the expense of a more complete dataset containing longer sequences (that could easily be handled with simple workarounds, such as chunking, or with other models allowing longer inputs).
* In Section 4.2, a more detailed explanation regarding the formulation of a text classification task using a generative model is expected. Moreover, the authors refer to the T5 paper while the task formulation is necessarily different between an encoder-decoder and their decoder-only model.
* In Section 4.3, the authors forget to mention which metrics are used for summarization (namely, Rouge).
* The five first sentences of the last paragraph in Section 5 are a repetition of what has already been explained in Section 3.2.

**Clarity:**

The paper is generally well written. There are a few typos in the paper though (e.g., "archtiecture" and "gits" in Section 5).

**Correctness:**

The dataset construction seems correct but is a bit under-detailed, as stated above.

**Documentation:**

The dataset is publicly available under the CC BY-NC-ND 4.0 license. The code is provided on Github with detailed steps to load the datasets, load the pre-trained model, and fine-tune it to reproduce the experimental results. The authors also provide a complete datasheet for the new dataset in the Appendix.

**Ethics:**

Some ethical concerns are mentioned in the "Ethical Considerations" section. No other salient concerns come to mind.

**Relation To Prior Work:**

* There seems to be no discussion of prior work in the paper, except for a brief description of the broader field of legalNLP in the Introduction section. The paper would benefit from an extensive discussion on other legal-domain benchmarks — such as LexGLUE (Chalkidis et al., 2021), FairLex (Chalkidis et al., 2022), JEC-QA (Zhong et al. 2019), CJRC (Duan et al., 2019), Swiss-Judgment-Prediction (Niklaus et al., 2021) — in order to help the reader better understand how this dataset may relate to the other benchmarks.
* Additionally, the authors could cite two valuable resources related to legal information retrieval in French and Japanese, respectively: (i) "A Statutory Article Retrieval Dataset in French" (https://aclanthology.org/2022.acl-long.468/); (ii) "COLIEE 2020: Methods for Legal Document Retrieval and Entailment" (https://link.springer.com/chapter/10.1007/978-3-030-79942-7_13).

**Summary And Contributions:**

The authors present a new benchmark for three different types of legal tasks (classification, judgement prediction, and summarization) in Korean. Specifically, they introduce six datasets: (1) a legal precedent corpus consisting of 150k precedents; (2) a multi-class legal text classification dataset consisting of 10,000 facts-category pairs with 100 different case categories; (3) another multi-class legal text classification dataset consisting of 2,760 facts-statutes pairs; (4) a legal judgement prediction dataset consisting of 11,000 cases from 7 categories; (5) another legal judgement prediction dataset consisting of 4,678 pairs of facts and degree of money claim acceptance; and (6) a summarization dataset consisting of 20,000 precedent-summary pairs. They also released a GPT2-like model for the Korean legal language and fine-tuned it on the different datasets.

---

> ### Author Response · Authors · 2022-08-24
> **Response to the reviewer 9YQQ**
>
> We thank the reviewer for the careful reading of the paper and the suggestions including valuable references. We'll make sure to improve the clarity and readability, and to include. missing references in the revised version of the paper. Below, we provide our replies to the major comments.
>
> [(2-1) How is the ground truth of the summarization dataset prepared?]\
> Please refer to our reply
> "[(2-1) How is the ground truth of the summarization dataset prepared?]" above.
>
> [(2-2) Examples with long textual inputs  in the summarization task]\
> We initially restricted the maximum input length to make the task more approachable. Upon the feedback, we have included additional examples with long input texts to make users exploit the full potential of the dataset. In the upgraded dataset there are a total of 40,892 training, 5,111 validation, and 5,111 test examples. The average number of tokens in the input text changes from 527 to 1,516 and of the summary becomes from 133  to  248. The maximum number of tokens in the input texts and the summaries are  93,420 and 6,536 respectively. The more details will be available in the revised paper. The resulting dataset shall be included in LBox Open soon.
>
> [(A-1) Motivation for the individual tasks]\
> From an academic view point, all tasks in general can be used to measure the performance of AI in legal natural language processing/understanding tasks.
>
> The case name classifier can be used to automatically recommend lawyers with proper background for  given factual description of cases. In civil cases, it is important to claim the rightful amount of money for optimal legal process. LJP-civil task mimics this situation and the trained model on this task can be used by legal practitioners as a research assistant tool.
>
> The models trained on the statute classification and LJP-criminal tasks can be used by laypersons to easily access the legal information. For instance, one can write the factual description and use the models to find related statutes and possible outcomes.
>
>
> [About effectiveness of Precedent corpus and the result of domain adaptation experiments]\
> We are thankful for the valuable feedback. We performed extensive domain adaptation experiments with KoGPT2, LCube-base, LCube-medium, and mt5-small. Here, the domain adaptation indicates the pre-training with the Precedent corpus only starting from the already pre-trained models on the general domain. We found that in the case name classification, the statute classification, and the Summarization tasks, the domain adaptation always leads to the improved performance. However we find in the legal judgement (LJP) tasks the domain adaptation does not show clear benefit. Especially in LJP-civil task, the domain adaptation always decreases the performance of GPT type models and LCube-base, and LCube-medium, the models trained from scratch show higher performance. This is consistent with previous findings that for difficult tasks, pre-training from scratch is more helpful than domain adaptation [Chalkidis, 2020, Legal-BERT]. The full experimental description and the result will be available soon on the revised paper.
>
> [(2-7) Extraction of the amount of money for LJP-civil task]\
> We train LCube with a prompt-tuning method to extract the information. We first prepared 160 training examples consisting of the ruling (or gist of claim) and parse pairs. The parse consists of the money provider, the money receiver, the amount of money, and the litigation cost. The ruling of criminal cases were parsed similarly. We are currently preparing a separate technical report explaining the details of the method.
>
> [The detail of the data engineering pipeline]\
> Please refer to our reply "[(2-5) The detail of the data engineering pipeline]" below.
>
>
>
> [The effects of label quantization on the LJP tasks]\
> Please refer to our reply "[(1-2), (3-3) The effects of label quantization on the LJP tasks]
> " above.

---

> > ### Comment · Reviewer_9YQQ · 2022-08-29
> > **Reply**
> >
> > The answers to my concerns are clear. Since most of the problems are fixed in the authors' revision, I will change the score from 5 to 7.

---

### Official Review · Reviewer_LvmP · 2022-07-28

**Rating:** 7
**Confidence:** 3

**Strengths:**

There are few pretraining corpora / datasets and legal-domain specific language models for Korean law. This paper introduces a large-scale Korean language legal corpus, different downstream NLP tasks relevant to the legal domain, such as legal judgment prediction, and a pretrained legal-domain language model, which I believe will be valuable in providing more multi-lingual NLP resources for the legal domain. Additionally, the dataset includes tasks over a diverse range of downstream tasks that are of interest to the legal NLP community, including more difficult tasks such as summarization that are often expensive to collect ground-truth labels for.

The authors put a significant amount of engineering effort and human validation into extracting structure data from raw data, since most Korean legal documents are available in either non-digitized document image or PDF formats.

Details were provided that would make it possible to reproduce pretraining for LCUBE-base and the fine-tuning procedure used for downstream tasks. I thought the experiments removing the facts from input were useful in showing the importance of the facts in the model reasoning about judgment predictions.

The data and language model are accessible through HuggingFace. See additional comments in Documentation section on potential improvements to accessibility and accountability.

**Weaknesses:**

I would have liked to see a little more justification about some of the quantization choices to form classification tasks out of the two legal judgment prediction tasks. It seemed like some different choices were made about maintaining class balance in quantization for the criminal vs. civil legal judgment prediction tasks and it was not entirely clear to me from the experiments whether the results would be affected by the somewhat subjective quantization decisions.

It might be helpful to discuss potentially reasons that the summarization task does not see improvements from legal domain-specific pretraining.

There is limited discussion of the social and ethical impacts of the dataset. I think two specific points that should be included are (1) the dataset’s license and whether any of the incorporated data from other sources is under other licenses, (2) whether there is potentially personal identifiable information in the legal case corpus and the norms / privacy protections governing Korean legal data released by the government and legal system.

**Additional Feedback:**

My main points of feedback are:
- More thorough justification or experimentation on the effects of label quantization counterfactuals for the legal judgment prediction tasks, which seemed somewhat subjective
- More detailed discussion of the licensing / distribution / use restrictions and if / where there may be personally identifiable information, which can in especially prevalent in some types of legal documents (and if so, elaboration on mitigation strategies or rationale for releasing dataset as is)


**Clarity:**

Overall, the paper was clear and easy to follow throughout.

I think the gains achieved from domain-specific pretraining (row 1 vs. row 2 in the table on page 7) can be pulled up and summarized more clearly earlier in the paper, in paragraph 3 on page 2.

There were a few small typos:
In Abstract, “the” missing right before “legal domain” in the third line.
Pg 2, “in last”->”in the last” in paragraph 2, line 5
Pg 5, Table 3, LJP-CIVIL subtask, “large-scalethe degrees”, replace with “the degree”?
Pg 5, “fine-tnue”->”fine-tune” in 4.1 Model training, line 2
Pg 6, missing “the” before “following” in paragraph 1, line 1
Pg 6, “archtiecture”->”architecture” Results, paragraph 2, line 2
Pg 7, “gits”->”gists” in paragraph 2, line 2

**Correctness:**

In terms of experimentation, I thought the experiment choices were reasonable and illustrative, with the LCUBE-base domain-specific pretrained language model compared against majority class and a general domain pretrained Korean language model with a comparable GPT-2 architecture.

It seemed that while it is true that compared to the other tasks, the gain in performance from domain-specific pretraining was smaller for the legal judgment prediction tasks, it was unclear to me whether this can be completely attributed to legal judgment tasks being inherently more challenging, as described in the second paragraph of the Results section. It appears that the gain from domain-specific pretraining is similar for LJP-CIVIL and CASE NAME, but the gain for LJP-CRIMINAL is much smaller. Could this be due to the number of classes being greater / the quantization being more granular or the classes being more balanced for the criminal LJP tasks? The share of LJP-CIVIL examples with label 2 is so small that the task simplifies somewhat to something close to a binary classification task, while the LJP-CRIMINAL tasks were quantized with consideration for both class balance and context specific legal aspects. I would have liked to see more justification for this or perhaps some experimentation that shows the somewhat subjective / human determined quantization doesn’t affect results significantly. What happens when the LJP tasks are modeled as regression tasks? Even if some of the predictive factors are not available (e.g., age), so performance might be expected to be worse, do the same relative differences in gains still hold?

**Documentation:**

I would have liked to see a more detail about how the ground truth summaries were collected for the summarization task. Was it crowdsourced?

The dataset and the pretrained language model are hosted on HuggingFace and I was able to access them there. A more detailed data card on HuggingFace could help more clearly document the intended use cases for the data and potential social impacts, biases, or limitations, as well as licensing information for the dataset. See comments in the Weaknesses section for specific points to address regarding impacts.

**Ethics:**

There was limited discussion of whether the dataset might contain personally identifiable information or any licensing / distribution / use restrictions on the dataset. Since the dataset was sourced from legal cases, there could be individuals’ names, but it is unclear whether this information has already been redacted by the institutions that initially released it.

**Relation To Prior Work:**

I think the methodology is similar to those of other recent English language legal NLP papers, but the paper clearly outlines that it differs in that it is providing Korean language datasets and pretrained language models for the legal domain, which has not been done at this scale in other work.

**Summary And Contributions:**

This paper introduces a Korean language legal NLP dataset, LBOX OPEN, and a Korean legal language model trained on LBOX OPEN, LCUBE, a decoder only model based on a GPT-2 architecture. The dataset includes a legal precedent corpus, two multi-class classification tasks, two legal judgment prediction tasks (with quantized ground-truth predictions) covering criminal and civil cases, and one legal summarization task.

The authors show that pretraining on a domain specific corpus result in performance gains across several of the tasks and comparable performance to a model with more capacity. They find that the domain specific pretraining is less helpful for the legal summarization task. These results are consistent with existing results on English language legal datasets and pretraining corpora. The originality comes primarily from the creation and evaluation of legal datasets and pretraining corpora in Korean.

---

> ### Author Response · Authors · 2022-08-24
> **Response to the reviewer LvmP**
>
> We thank the reviewer for providing valuable feedback while recognizing the novelty of the dataset. Below we provide our answer to the major comments.
>
>
> [(1-2), (3-3) The effects of label quantization on the LJP tasks]\
> To investigate how the model performance changes under different quantization schemes,  we performed (1) further research on how the Korean courts decide sentences in law, and (2) additional experiments under different quantization or regression settings.
>
> 1. In our initial study, we quantize the output spaces of LJP-civil, and LJP-criminal based on the subset of knowledge of the Korean legal system. A further study reveals that the sentencing commission of the Korean government provides the complex baseline for individual case categories (https://sc.scourt.go.kr/sc/engsc/ebook/index.html#page=1). Although there are some notable features like "above 1 year, the min and max imprisonment period increased by at least by a half year", it was difficult to find common features that can be used for the quantization. Thus instead of trying to find legally grounded quantization boundaries, we extend the experiment on LJP tasks in two directions, (1) regression task, and (2) the new quantization scheme that balances the label distribution of the datasets.
> 2. In LJP-civil tasks, we perform additional experiments by quantizing the claim acceptance ratio with 0.1 interval mimicking the regression setting (when the range is quantized uniformly and their interval becomes infinitesimal, the classification task becomes a regression task [Bishop, 2006]). We also measure the performance of several new models (KoGPT2 w/ domain adaptation, LCube w/o legal corpus, LCube-medium (124M -> 354M), mt5-small w/ domain adaptation, mt5-large). Here, the domain adaptation indicates the pre-training with only the legal corpus starting from the already pre-trained models on the general domain. We find in both the regression and the original task setting, use of legal corpus during the pre-training stage increases the model performance whereas domain adaptation decreases the performance across all models except mt5s. On the contrary, in the case name and statute classification tasks, the domain adaptation always increases the model performance. This is in line with the result from the previous study that for difficult tasks, pre-training from scratch is more helpful than domain adaptation [Chalkidis, 2020, Legal-BERT]. Also in LJP-criminal tasks, we consistently find across all models (1) the removal of facts decreases the performance, (2) inclusion of the ``the reason for sentencing'' increase the performance, and (3) the performance drop as the the quantization interval becomes smaller (as the task becomes regression task). Unfortunately, unlike other tasks, we could not find clear winning models for LJP-criminal task.
>
>
> [(3-1) More discussion on the social and ethical impacts]\
> The Korean precedents are officially not protected by the copyright law (https://law.go.kr/법령/저작권법/제7조).
> For the redaction process, please see
> "[(3-1)The redaction process for the precedent corpus]" above.
>
>
> [(2-1) How is the ground truth of the summarization dataset prepared?]\
> The ground truth of the summarization tasks is extracted from the Korean Supreme Decision report that contains a summary of decisions written by Director of Judicial Research from Supreme Court Library of Korea (https://www.scourt.go.kr/eng/supreme/decisions/guide.jsp, https://www.scourt.go.kr/portal/gongbo/PeoplePopupView.work?gubun=24&seqNum=1827)
>
> [(1-3) More experiments and analysis on the summarization task]\
> In the summarization task, LCube-base shows the lower score compared to KoGPT2 and mt5-small. Individual analysis reveals, LCube generates a shorter summary (the mean number of characters n = 195) compared to KoGPT2 (n=277) and mt5-small (n=299). According to a recent study, the training corpus significantly affects the downstream performance of language models [Shin et al., NAACL, 2022]] and this may account for the result. Additionally we also perform extra domain adaptation and scaling experiments where we find both factors increase the rudge scores. Also, we found that the stemmer used for computing the rouge scores does not support Korean and thus re-calculate the rudge-scores at word level. The full experimental description and the result will be available soon on the revised paper.
>
> - Chalkidis, 2020, LEGAL-BERT: The Muppets straight out of Law School
> - Shin et al., On the Effect of Pretraining Corpora on In-context Learning by a Large-scale Language Model, 2022

---

> > ### Comment · Reviewer_LvmP · 2022-09-03
> > **Reply**
> >
> > The comments and ethic review response addressed my central questions. I've updated my score to 7.

---

### Review · Ethics_Reviewer_oMd1 · 2022-08-22

**Recommendation:** 2

**Ethics Documentation:**

They describe where they got their source data from, but not how the original data was licensed. In terms of ethical and responsible use, see the two notes above.

**Ethics Review:**

The paper discusses one ethical concern, but fails to consider two others that I believe are more important:

1. The potential that the data contains unredacted sensitive information about individuals, including identifying information such as full name along with things like alleged crimes.  Having one's name linked to an alleged crime can have harmful impacts for people, including harms to reputation as well as potential employment and housing. The work done to extract structured data from these documents makes this type of information more accessible, including to search engines, and the authors do not discuss whether the data contains this type of personal information and if so what steps they are taking to address the potential harms. (Note: I see in the checklist section at the end that the authors have reviewed the data for personally identifiable information, some details about what they did in the paper itself would help)

2. A downstream use-case for a training data set such as this one is to train risk assessments and related automated decision-making tools deployed by the courts and prosecutors. As reviewer 7fpy mentions in [their review](https://openreview.net/forum?id=TaARsI_Iio&noteId=2Mt-XLLkAPV), such tools have, in other contexts, had the effect of reproducing or amplifying existing bias and social inequality. The authors should address whether this data set is appropriate for such use-cases, and also attempt to look at the data they've collected through the lens of bias and discrimination: are some groups in this data over or under-represented or represented in such a way that predictive models trained on this data would have disproportionately negative impact on them?

---

> ### Author Response · Authors · 2022-08-24
> **Response to the ethics reviewer oMd1**
>
> We thank the reviewer for raising a crucial concern. In short, while the data construction process strictly adheres to license regulations and minimizes privacy-related issues, it seems we have not sufficiently discussed them in the paper.  Below we provide the details, and we will make sure to include them in the next revision.
>
> [The license of Korean precedents]\
> The Korean precedents are not protected by the copyright law (COPYRIGHT ACT Acticle 7, https://law.go.kr/LSW/lsInfoP.do?lsiSeq=192474&viewCls=engLsInfoR&urlMode=engLsInfoR#0000, https://law.go.kr/법령/저작권법/제7조).
>
> [(3-1)The redaction process for the precedent corpus]\
> The precedent corpus consists of ~80k publicly available precedents from the government of Korea and ~70k precedents from our internal database. For all precedents, personal information is redacted by the government before the initial release following the official protocol (https://www.scourt.go.kr/portal/information/finalruling/anony/index.html,https://glaw.scourt.go.kr/wsjo/gchick/sjo330.do?contId=3202812#1660965241959).
>
> Data subjected to anonymization are as follows:
> - Name and the equivalents: Name, nickname, pen name, ID, and corresponding nouns that point to a specific person are replaced with upper case alphabets A, B, C, etc., without redundancy.
> - Contact information: Phone number, e-mail address, residential address and corresponding contact data are deleted in the meta information(당사자 단락) and replaced with upper case alphabets in the reasoning(이유 단락)
> -  Financial information: Account number, credit card number, check number and corresponding financial data are replaced with upper case alphabets without redundancy.
> -  Other personally identifiable information: Social security number is deleted. car registration number, address of real estate holdings and corresponding personal information are replaced with upper case alphabets without redundancy.
>
> In the case of felonies such as serial killing, there are some exceptions where the Korean government can decide not to anonymize for the social benefits. For example, see https://en.wikipedia.org/wiki/Yoo_Young-chul, https://www.law.go.kr/판례/(2004고합972).
>
> In addition to this, for ~70k precedents from our internal database, we follow our additional internal redaction process; (1) apply NER module in search of redaction candidates; (2) the employs of our internal data labeling platform LWorks (https://lworks.kr/work) examine the model  prediction results;
>
> [2. The limitation in using the dataset for automatic legal decision making]\
> The models trained on the legal judgement prediction (LJP) datasets can be used for automatic legal decisions and like any data-driven approaches, the models can predict biased results depending on the data statistics of the training set.
> To minimize the risk, we have used only the precedents redacted by the Korean government following the official protocol (see above). Also, we have fully open-sourced the dataset, trained model, and training code so that anyone can access and investigate the dataset and the method without any restriction.
>
> Nevertheless as the population of precedents is already biased reflecting the current status of the society (for example, the judicial yearbook published in 2021 reveals the 86.8% defendants in 1st criminal trials were male ((https://www.scourt.go.kr/img/pub/jur_2021_Book6.pdf)), and as we randomly sampled the population during the construction of LBox Open whenever possible, the use of the released dataset and the model without explicitly understanding the potential issue from the dataset bias can make negative social impacts.
>
> To concretely address this issue, we performed a case study about gender bias in the criminal precedents (race, language, and regions of litigants are not considered as our datasets consist of Korean cases only). The age is also not considered at this time as judges can reduce or increase the sentences based on the ages of litigants following the official sentencing guidelines.
>
> We first extracted the victims' gender information from the facts using regular expressions. Among five criminal case categories in the LJP-criminal dataset, only “indecent act by compulsion” (강제추행) shows strong bias in victims' gender (1,451 females, 74 males). We examined how the prediction of LCube changes when the victims' genders are exchanged (as only 0.46% cases include the gender of the defendant in the facts, we change the gender of the victims). Among 131 test examples, the prediction changed in 11 examples. Further analysis revealed in 10 examples (7.6%) the outcome sentences decreased when the victims' genders were changed from female to male. Although this is relatively small (91.5% outcomes remain identical), we'll update the result to the paper to clarify the potential risk and the limitation of automatic legal decision making based on our dataset. Finally we would like to remark that there is also a social benefit in democratizing the data.

---

> ### Comment · Reviewer_7Vi4 · 2022-09-01
> **My issues have been addressed**
>
> I can't find a way to update/change my ethics review (it's only letting me access the "Official Review" screen and not the Ethics Review one, but I am Ethics Reviewer oMd1 and the authors have addressed my concerns with their recent updates. Thank you!

---

### Author Response · Authors · 2022-08-24
**Response to all reviewers**


We sincerely thank all the reviewers for their interest in our work and providing many valuable feedbacks. While all the reviewers recognize our first novel effort to build large-scale Korean legal AI dataset and language models from the raw texts and document images, they also (1) suggest to strengthen the experimental part, (2) request more explanation on our data engineering pipeline and the dataset construction process, and (3) raise privacy and license-related concerns.

In response to this we have included

(1-1) new scaling and domain adaptation experiments with both GPT and T5 by using at least 9 different model checkpoints in the fine-tuning experiments on the individual legal AI tasks.

(1-2) new legal judgement prediction (LJP) experiments under regression or different range quantization schemes.

(1-3) new analysis and experiments on the summarization task.

\
(2-1) how the GT of the summarization dataset were prepared

(2-2) new 31,114 examples with long input  text in the summarization task

(2-3) new 19,989 examples in the case name classification task extending case categories from 100 to 603.

(2-4) new 14,432 examples in the statute classification task extending the number of statute categories from 169 to 1,015 and case categories from 46 to 434.

(2-5) the additional explanation about the ML modules in our data engineering pipelines and their individual performances and the overall automation ratio in the deployed system.

(2-6) the brief description of how we have prepared the ML modules

(2-7) the brief description about how we develop custom language model to parse rulings

\
(3-1) the references and the brief description of how Korean government removes personal information

(3-2) the practical difficulty in getting access to the raw data in scale

(3-3) the official references of the sentencing guideline (related to LJP tasks)
\

Below we provide our answers to the individual comments. The full details will be available on Aug 26 from the next revision.
To go to the detail of our response on individual topics above, one can search "(#-#)". For example, to see our reply about the new data statistics of the summarization dataset, search "(2-2)".

---

### Author Response · Authors · 2022-08-26
**The paper updated.**

Dear reviewers

Sorry for being late.
We have updated the paper with

(1) new experimental results (Table 3,4 in page 8; Table 5 in p9, Table 10 in p19)

(2) new data statistics (Table 1 in p4,  Table 2 in p5, Fig. 2 in p17, Table 6, Table 7 in p18)

Also

(3) the result section is entirely re-written (pp. 7--10);

(4) the ethical considerations section is significantly updated (p10);

(5) the explanation about the license of  Korean precedents and the official redaction protocol is added (Sec. 2.2);

(6) the explanation of the precedent disclosure status has included (Sec. 2.2);

(7) the explanation about the new datasets are added (pp.5-6);

(8) the more explanation about the data engineering pipeline and the automation efficiency is included in Appendix 3 (p17).

(9) other miscellaneous updates are indicated by red colors.

~The "related works" section will be updated before next Monday.~

(10) The "related works" section is updated (Appendix A.1)


## Final remark
Under the current Korean legal system, accessing the precedents costs 1,000 KRW per case.
We would like to emphasize that, as the released corpus includes the 65k precedents from our internal database, the cost of making the corpus is at least 65,000,000 KRW ($\sim$50,000 USD) just for purchasing the raw data not to mention the data engineering cost.

Furthermore, we have released the entire datasets anonymized following the official redaction protocol of Korean government, and the models with their training code.
This will enable the scientific approaches and debates on various legal AI topics including social impact of employing ML-based legal judgement prediction.

We hope our replies and the revision cover all the concerns from the reviewers comprehensively.

---

### Meta-Review · Area_Chair_cMSc · 2022-09-08

**Recommendation:** Accept
**Confidence:** 4

**Metareview:**

This paper presents a large corpus of Korean legal documents, paired with labels corresponding to two classification tasks, two legal judgment prediction tasks, and one summarization task. Reviewers praised the uniqueness of the new resource.

However, there is some criticism of the results section, with several reviewers bringing up possible comparisons to other models and wondering whether the performance of this model has been analyzed thoroughly.  The authors argue in particular that mT5 and KoGPT-2 can benefit from continued pre-training this corpus, which seems satisfactory, although there is the question raised by 7fpy that this is essentially pre-training on the test tasks and may be too generous when assessing the performance on other legal-domain tasks that arise down the road.  Reviewer 7fpy brings up some valid critiques of the dataset construction: several points about filtering were unclear, as was the strength of the automatic pipeline used to construct the dataset.

There are a few ethical concerns raised by the ethics reviewer, which I believe are serious. Point [1] is resolved. Point [2] about misuse seems valid and harder to argue against.  The data does have value for use in the kind of more benign studies the authors report in the response there. But its very existence could encourage the construction of certain kinds of automated tools for legal judgment prediction, which has been a hotly-debated issue in the NLP community before. Ultimately I will defer to the ethics reviewer on this one, who seems satisfied.

Overall this paper seems like a well-done effort. The main question is whether its utility (modulated by issues with the results) outweighs the risks of putting it out there. Reviewers seem to lean positive on this and I would lean positive as well.

---

### Decision · Program_Chairs · 2022-09-16

Accept